# Mineral and Vitamin Intakes of Latvian Women during Lactation Period [note 1]

**DOI:** 10.3390/foods11030259

**Published:** 2022-01-19

**Authors:** Līva Aumeistere, Alīna Beluško, Inga Ciproviča, Dace Zavadska

**Affiliations:** 1Faculty of Food Technology, Latvia University of Life Sciences and Technologies, Rīgas iela 22a, LV-3004 Jelgava, Latvia; alinapolipartova@gmail.com (A.B.); inga.ciprovica@llu.lv (I.C.); 2Department of Pediatrics, Riga Stradiņš University, Vienības gatve 45, LV-1004 Riga, Latvia; dace.zavadska@rsu.lv

**Keywords:** human milk, lactation, diet, micronutrients, vitamins, minerals

## Abstract

Studies from Europe and the United States indicate that women during the lactation period do not consume sufficient amounts of essential micronutrients. Previously reported data from Latvia indicates a low vegetable, fruit, fish, cereal, and milk and dairy products intake among lactating women. This raises concerns that nutrient (especially minerals and vitamins) intakes could also be insufficient. Therefore, this study aimed to assess mineral and vitamin intakes among lactating women in Latvia in comparison to nutritional guidelines at both a national and European level. 72-h food diaries were collected from 62 participants during the period November 2016 till December 2017 and from 68 participants during the period from January 2020 to December 2020. This also allowed us to evaluate whether nutrient intakes among lactating women in Latvia have changed in recent years. The Fineli Food Composition Database was used to calculate micronutrient intakes among the participants. MS Excel 2019 and IBM SPSS Statistics 23 were used for the statistical data analysis. The results revealed that dietary intakes of calcium, iron, iodine, and vitamins A, D, B_1_, and B_9_ among the participants of both study periods did not meet dietary recommendations. Low mineral and vitamin intakes could potentially affect the composition of human milk, and therefore micronutrient intakes, for breastfed infants. This indicates a need to develop dietary guidelines in order to improve diets among lactating women in Latvia.

## 1. Introduction

For approximately 20% of the infants in Latvia, human milk is the sole nutrient source in the first six months of life [1]. Therefore, it is important that human milk contains a sufficient amount of nutrients required for the adequate growth and development of the infant [2]. Nutrients for human milk are derived from maternal nutrition [2]. Therefore, a healthy diverse diet and sufficient nutrient intake for women during the lactation period are essential [3,4].

Overall, it has been established that the macronutrient (total fat, protein, lactose) content in human milk is not influenced by the maternal diet [5]. However, the mineral and vitamin content in human milk can be directly affected by the maternal intake of these micronutrients [5]. Therefore, it is essential that women during lactation period consume a well-balanced and diverse diet to meet increased nutritional needs and to ensure a sufficient number of micronutrients for the breastfed infant via human milk [4].

There are many studies that have evaluated the diet of women during the gestation period [6,7,8,9]. Fewer data are available regarding the diet of women during the lactation period (and especially for women in recent years) [8,9]. Studies which have evaluated the diet of lactating women indicate that women during postpartum period are primarily focused on taking care of the infant, and thus pay less attention to the quality of their diet [10]. For example, Krešic et al. (2012) [11] reported that Croatian lactating women are deficient in magnesium, zinc, vitamins A, D, B_1_, B_6_, and B_9_. Researchers from Spain [10] reported that during the first month postpartum, women were deficient in vitamins A, D, B_9_, and potassium. Pratt et al. (2014) [12] advised that lactating women in the United States should increase their intake of vitamins A, D, E, and C and iodine, which may be accomplished by educating lactating women about the importance of consumption of vegetables, fruits, berries, dairy products, and fatty fish.

Other researches [13] have also raised concerns that many women eliminate certain foods from their diet during lactation period to avoid a possible situation where the breastfed infant might present reactions to the allergenic food components found in human milk. Many women still believe that infantile colic might be caused by the consumption of legumes, cabbage, onion, garlic, sparkling beverages, etc., and therefore avoid these products during the lactation period [13]. However, the evidence of the effectiveness of dietary modifications for the treatment of infantile colic are still inconclusive [14]. Therefore, it should be more emphasized that food restrictions during the lactation period can be detrimental for mothers’ health [13].

Despite the importance of maternal nutrition during the lactation period, in some countries (for example, Spain and Latvia) there are no specific dietary guidelines for this period, which could be due to the lack of studies evaluating women’s nutritional status during the lactation period compared to those during gestation [10].

It is necessary to develop dietary guidelines for lactating women in Latvia since previously reported data from Latvia indicate a low vegetable, fruit, fish, cereal, and milk and dairy products intake among women during the lactation period [15]. It is already established that women during the lactation period in Latvia consume less energy, carbohydrates, eicosapentaenoic, and docosahexaenoic acid than recommended [15]. However, data regarding micronutrients intake among women during the lactation period in Latvia is limited. Therefore, this study aimed to assess whether vitamin and mineral intakes among the study participants were sufficient to meet the daily nutritional recommendations set for lactating women at both a national and European level.

## 2. Materials and Methods

This cross-sectional study was performed in two intervals, allowing us to compare the diet of lactating women in Latvia within recent years. Women were invited to participate in the study via a poster published on social media. To calculate the minimum sample size for the study period, an online calculator was used [16]. With a probability of 0.05 and a confidence level of 95%, it was calculated that at least 59 participants were needed in each study period.

Initially, 212 participants met the study inclusion criteria, but 82 women decided not to participate after getting acquainted with the instructions of the study or were unable to collect a sufficient amount of human milk for the study (data regarding human milk composition among lactating women in Latvia are reported in articles Aumeistere et al., 2019 [15] and Aumeistere et al., 2021 [17]). Overall, 44 participants dropped out from the first study period and 38 participants from the second study period. In the end, 130 lactating women participated in this study:62 participants during the period November 2016 till December 2017 and68 participants from January 2020 to December 2020.

Each woman participated only once and only in one of the study periods.

Prior to the study, the approval from the Riga Stradinš University Ethics Committee was obtained in the year 2016 (protocol code 4/28.7.2016.) and renewed in the year 2020 (protocol code 6-1/01/6). Written informed consent was obtained from all of the participants before the study.

The inclusion criteria for participants were:Signed consent blank;Reside in Latvia;Singleton pregnancy;At least one month postpartum;Exclusively or partially breastfeeding;Participating women and her child apparently healthy (without metabolic disorders, no acute illnesses, etc.).

During the study, participants had to choose three consecutive days to fulfil the 72-h food diary. No dietary restrictions were defined, and participants were able to consume a self-chosen diet (no dietary restrictions were applied). Volume measures (handful, teaspoon etc.) were allowed to be used to help complete the food diary. Vitamin and mineral intakes were calculated using the Fineli Food Composition Database [18]. If participants were using dietary supplements during the study period, the nutritional information was taken from the manufacturers’ websites and included in the calculations of total micronutrient intakes.

Information about characteristics such as maternal age, parity, time postpartum, sex and birth weight & length of the infant, and breastfeeding pattern (exclusive or partial) was also collected.

Food diary data was exported from Fineli Food Composition Database to the MS Excel 2019. IBM SPSS Statistics 23 was used for data statistical analysis. The data were expressed as mean ± standard deviation or median [interquartile range] and minimal–maximal values. Non-parametrical statistical tests such as the Kruskal-Wallis test and Spearman correlation were applied to analyse the data. A *p*-value of ≤0.05 was considered statistically significant.

## 3. Results

### 3.1. Characteristics of the Participants

Mean value ± standard deviation, as well as minimal–maximal values for the characteristics of the participants, are compiled in Table 1. No statistically significant differences were found comparing both study groups (*p* > 0.05). According to the body mass index calculations, the majority of the participants had a normal weight (*n* = 101), five participants were underweight, nineteen participants were overweight, and five were obese.

Fourteen participants from the first study period and ten participants from the second study period in the 72-h food diary noted that they avoid milk and dairy products (Table 2). The reasons for the exclusion were as follows: participant was a vegetarian (two from the first study period, one from the second study period) or a vegan (two participants in each study period), maternal health issues (two participants from the first study period), and the infant had cow’s milk protein allergy (eight participants from the first study period and seven participants from the second study period).

Other food groups excluded from the diet among the study participants are also compiled in Table 2.

### 3.2. Mineral and Vitamin Intakes among the Participants

Mineral intakes among the participants in comparison to recommended intakes are compiled in Table 3, and vitamin intakes are included in Table 4.

No significant differences were found regarding mineral and vitamin intakes comparing both study periods (*p* > 0.05), except for vitamin D, which was higher among the participants from the second study period (*p* = 0.02).

Overall, median phosphorus, magnesium, vitamins K, B_3_, B_6_, and B_12_ intake among the participants was sufficient (the recommended or adequate intake was reached for at least 75% of the participants) (Figure 1).

Participants were consuming less calcium than required, and for only 42% of the participants, the median calcium intake reached the recommended intake of 900 mg per day (Figure 1). Similarly, only for approximately 40% of the participants, the median iron intake reached the recommended intake of 15 mg per day (Figure 1). Slightly lower than recommended was the zinc intake, reaching 11 mg of zinc per day for only 55% of the participants (Figure 1). Iodine intake was also low. The iodine intake reached at least 200 μg per day for approximately 30% of the participants (Figure 1).

On the contrary, sodium intake was above the recommended intake of 2000 mg per day for the majority of the participants (Table 3 and Figure 1).

The intake of many vitamins (A, D, B_1_, B_2_, B_9_) among the most participants did not reach the recommended intake for lactating women (Table 4). For example, for only 30% of the participants, the median vitamin A intake reached the recommended intake of 1100 μg of retinol activity equivalents (Figure 1). Similar results were obtained regarding vitamin D intake. Furthermore, only around 30% of participants consumed the recommended intake of vitamin D (10 μg per day). However, for 8% of the participants, the daily vitamin D intake also exceeded the tolerable upper intake level of 100 μg (Figure 1). There was especially low data reported regarding vitamin B_9_ intake. For ~40% of the participants, the median vitamin B_9_ intake did not even reach half of the recommended intake (Figure 1).

The majority of the participants from both study periods noted the use of at least one dietary supplement (32 participants from the first study period and 43 participants from the second study period). Information regarding nutrient intakes via dietary supplements are compiled in Table 5. Vitamin D was the most often consumed dietary supplement in both study periods (*n* = 56), followed by vitamin C (*n* = 32), iron (*n* = 31) and vitamin B_9_ (*n* = 31) (Table 5). Overall, dietary supplement consumption was higher among the study participants in the second study period. In particular, a more frequent consumption was reported regarding vitamin D, zinc, magnesium, and vitamin E intake (Table 5).

## 4. Discussion

A diverse diet is essential to meet the requirements of nutrient intake during the lactation period. Foodstuff from different food groups such as cereals, potatoes, milk and dairy products, meat, fish, vegetables, fruits, berries, and plant-based fats (e.g., nuts, seeds, vegetable oils) should be included in every-day diet [20]. However, previously conducted research in Latvia indicates that women during the lactation period have an inadequate intake of vegetables, fruits, berries, cereal products, milk and dairy products, and fish [15]. Therefore, it is not surprising that many individual mineral and vitamin intakes (calcium, iron, iodine, vitamins A, D, B_1_, and B_9_) among the participants of this study were also low.

Maternal calcium intake will not affect the calcium content in human milk due to the fact that a woman’s body adapts to the additional nutrient loss via human milk (e.g., mobilisation of calcium from body stores, etc.) [22]. Nevertheless, calcium intake during the lactation period should reach 900 mg per day [19]. Not only this research, but also studies from Spain [8], Poland [23] and France [24] have reported that calcium intake among lactating women is low. In this study, the majority (~60%) of the participants’ median calcium intake did not reach the recommended 900 mg per day. This could be related to the fact that many participants were excluding milk and dairy products from their diet (based on the information noted in the 72-h food diaries), which are a good source of calcium [20]. Women should be informed that the primary source of extra calcium in human milk becomes the maternal bones which leads to bone mineral density loss during lactation period, and therefore sufficient calcium intake is vital [22].

Although the phosphorus content in human milk is not affected by the maternal intake of phosphorus [2], to compensate for the loss of this micronutrient via human milk, women during lactation are advised to increase their intake of phosphorus up to 900 mg per day [19]. Phosphorus is widely distributed in different food products (both animal- and plant-based) [20]. This could explain why phosphorus intake reached the recommend intake among the majority of the study participants. Also, other studies point that phosphorus intake among lactating women reaches the recommended intake and usually exceeds 1000 mg per day [11,12,23,24].

The content of potassium is not affected by maternal diet [2]. It is recommended that lactating women should consume 3100 mg of potassium per day [19]. Although potassium can be easily consumed by including plant-based food products (fresh and dried fruits, legumes, green leafy vegetables, etc.) in their diet [20], studies from the United States [12] and Poland [23] have report potassium intake among lactating women to be lower than that recommended by the national nutrition guidelines. Although previous data indicate that potassium rich sources such as vegetables, legumes, fruits and berries are consumed on daily bases by only small percentage of lactating women in Latvia [15], the calculated potassium intake for more than half of the study participants reached the recommended intake of 3100 mg per day.

In contrast to other nutrients, sodium intake should be limited to an amount of 2000 mg per day since high sodium intake is associated with higher blood pressure and water retention in the body [19,25]. A previously conducted study in Latvia already showed that Latvian adults exceed the recommended amount of sodium, and the dominant source of sodium in their diet is processed foodstuff [25]. Although previous data show that sodium-rich sources such as salty snacks and fast foods among lactating women in Latvia are consumed seldomly [15], according to the evaluation of 72-h food diaries, sodium intake among the participants of this study was higher than recommended. Higher sodium intake among the study participants could be due to meat products such as sausages, ham, etc., and cheese consumption. These products usually contain a significant amount of salt (and therefore sodium), usually more than 1 g of salt per 100 g of the product [18]. Similar observations regarding a higher sodium intake among lactating women than recommended were also reported by Bzikowska-Jura et al. (2018) [23] and Pratt et al. (2014) [12].

During the lactation period, magnesium for human milk is mobilised from the body stores [26]. Therefore, the maternal intake of magnesium does not directly influence magnesium content in human milk [2]. Also, based on nutritional guidelines, requirements for magnesium during the lactation period are not increased (kept at 280 mg per day) [19]. The main sources of magnesium in the diet are cereals, legumes, nuts, and green leafy vegetables [20]. We report a higher magnesium intake among the lactating women compared to data from Croatia and United States [11,12], but this is probably related to the additional magnesium intake via dietary supplements that was reported among the study participants. This affected the total magnesium intake accordingly.

Meat, especially red meat, is a good source of iron [20]. However, frequent consumption of red meat increases the risk of colorectal cancer and coronary heart disease. Therefore, dietary guidelines suggest limiting red meat intake no more than 500 g per week and replacing it with poultry, fish, and legumes [20]. According to previously reported data from Latvia, meat and meat products among lactating women are mostly consumed only once to twice a week [15]. Also, it should be noted that few participants of this study were vegetarians or vegans. This could explain why median iron intake among the study participants did not reach the recommended intake of 15 mg per day. To increase iron intake among lactating women, red meat intake could be increased to two or three times per week (but not more than 500 g per week), and other iron-rich sources such as poultry, fish, and legumes should be included in the diet [20].

Even lower iron intake among lactating women (only around 8 mg per day) compared to results obtained in this study (around 13 mg per day) has been reported by researchers in Spain [8]. Researchers from Poland, the United States, France, and Croatia do not report insufficient iron intake among lactating women [11,12,23,24]. However, their national nutritional guidelines have also set a lower iron intake for women during the lactation period (9 mg per day according to Croatian and the United States, and 10 mg per day according to France’s and Poland’s nutritional guidelines) [11,23,24,27].

Although zinc content in human milk is not directly affected by the maternal intake of this micronutrient [2], the nutritional requirements for zinc during lactation period significantly increases (up to 11 mg per day) [19]. Median zinc intake among the study participants from the first study period was lower than recommended, but median intake among the second study period was within recommendations (11 mg per day). Lower zinc intake could be related to the fact that zinc-rich food sources such as meat, eggs, and fish are not consumed on daily basis among lactating women in Latvia [15]. Furthermore, researchers from Poland, France, Croatia report lower zinc intake among lactating women than recommended [11,23,24].

Selenium is one of the few micronutrients whose content in human milk is directly affected by the maternal diet [28]. Therefore, lactating women are encouraged to consume 60 μg of selenium per day [19]. Fish, meat, eggs, milk, and dairy products are a good source of selenium [20]. Selenium intake among the study participants was even a little higher than recommended (~80 to 90 μg per day, respectively). Similar values for selenium intake among lactating women have been reported from Croatia [11]. Fish and seafood are rarely consumed among lactating women in Latvia, and therefore the main sources for selenium in the diet are probably meat, meat products, and eggs [15]. However, it should be noted that the calculated values regarding selenium intake in our study were taken not only from food but also from dietary supplements.

Iodine content in human milk is also directly influenced by the maternal intake of iodine [5]. Therefore, lactating women should increase their iodine intake and consume 150 μg to 250 μg of iodine per day [19]. Iodine intake among the study participants reached the minimum recommended value of 150 μg per day. The minimal value of 150 μg per day is set in national Latvian nutritional guidelines [19], but other nutritional guidelines at the European level recommend a higher iodine intake during the lactation period (200 μg of iodine per day) [20,21]. Lower iodine intake values (only around 100 μg per day) have been reported from Poland and France [23,24]. Data from the United States reported even lower iodine intake among lactating women—only around 60 μg per day [12]. Nevertheless, it should be noted that sixteen participants of this study noted also the use of a dietary supplement containing iodine, which had an impact on the median iodine values reported. Overall iodine-rich food (such as fish and milk and dairy products) intake among lactating women in Latvia is low [15], which means that without additional iodine intake via dietary supplements, women are probably unavailable to fulfil increased iodine requirements during the lactation period.

The vitamin A content in human milk can be influenced by the maternal intake of vitamin A [26]. Vitamin A is widely found in both animal origin foods (dairy products such as butter, cheese, egg yolk, etc.) as well as in plant-based products as provitamin A (dark green leafy vegetables, red- and orange-coloured vegetables, fruits and berries, etc.) [20]. Nevertheless, median vitamin A intake among the participants did not reach the recommended intake of 1100 μg per day (expressed as retinol activity equivalents). This is probably due to the fact that vegetables, fruits, berries, and dairy products are not consumed on a daily basis by lactating women in Latvia [15]. Furthermore, other researchers have reported insufficient vitamin A intake among lactating women due to a low intake of vegetables, fruits and other vitamin A sources [11,12,24]. Therefore, lactating women should be advised to include in the diet more food products such as vegetables, fruits, berries, as well as animal origin products (such as milk, cheese, butter, eggs, etc.).

Vitamin E content in maternal diet is not associated with vitamin E content in human milk [26]. However, to compensate for additional vitamin E loss via human milk, women during the lactation period should increase their vitamin E intake up to 11 mg per day [19]. Good sources of vitamin E are vegetable oils, nuts, seeds, and egg yolk [20]. Median vitamin E intake among study participants in both study periods was sufficient (around 13 mg per day), and we report the highest vitamin E intake among lactating women compared to data from Poland, France and the United States [12,23,24]. Higher reported vitamin E intake in this study was probably related to the fact that many participants (*n* = 27) noted in the 72-h food diary that they use also a dietary supplement containing vitamin E.

Although some food products such as fatty fish, egg yolk, mushrooms contain vitamin D, the basic requirements for vitamin D can be satisfied by exposing skin to the sun [20]. Although human milk contains only traces of vitamin D (0.05 μg 100 mL^−1^) [2], to provide a sufficient supply of vitamin D and ensure woman’s requirements for vitamin D, additional vitamin D intake during pregnancy and lactation period via dietary supplements is recommended (10 μg of vitamin D per day, and 20 μg per day in winter) [29].

Although ~40% of the study participants noted that they use dietary supplements containing D vitamin, the median vitamin D intake among participating women was only around 5 to 6 μg per day, which is half of the recommended intake (10 μg per day). Overall, a higher D vitamin intake was reported among the participants from the second study group, probably due to the fact that D vitamin supplements consumption were also higher among the participants from this study group. It is hard to say why D vitamin intake via supplements was higher among the second study group. It is possible that health professionals in Latvia in recent years are more frequently advising women to pay extra attention to sufficient vitamin D intake during both the gestation and lactation period and recommending additional vitamin D intake. Nevertheless, obtained data of our study indicate that women in Latvia need to be further informed about the additional vitamin D intake via dietary supplements since it was not the case that all participants reached the recommended daily vitamin D intake. Researchers from Spain [8] report even lower vitamin D intake (~2 μg per day) among women in the post-partum period (only 12% of the recommended intake according to their nutritional guidelines (15 μg of vitamin D per day).

Nevertheless, it should be noted that also for 8% of the participants of this study, the median vitamin D intake exceeded the tolerable upper intake level of 100 μg per day. Chronic excess intake of vitamin D may lead to tissue calcification and result in renal and cardiovascular damage [30].

Vitamin K content in human milk is low (~0.25 μg 100 mL^−1^) and not affected by maternal intake of vitamin K [2,26]. According to nutritional guidelines, vitamin K intake during lactation should be kept 70 μg per day [21]. Green leafy vegetables and vegetable oils are the main source of vitamin K in the diet [20]. Although vegetables are not frequently consumed among lactating women in Latvia [15], according to calculations, median vitamin K intake among study participants was adequate (around 120 μg). Few participants (*n* = 9) noted that they consume vitamin K additionally with dietary supplements. Lower vitamin K intake (100 μg per day) among lactating women was reported by Pratt et al. (2014) [12].

Vitamin B_1_ content in human milk is directly affected by the maternal diet, and therefore women during the lactation period should increase their intake of vitamin B_1_ up to 1.6 mg per day [2,19,26]. Vitamin B_1_ sources in the diet are cereals and cereal products and meat and meat products [20]. However, previous data regarding habitual food intake among lactating women in Latvia report seldom consumption of cereals & cereal products and meat & meat products consumption—once, twice a week [15]. This could potentially explain why vitamin B_1_ intake among the participants was lower than recommended. Lower than recommended vitamin B_1_ intakes have been reported also among lactating women from Croatia [11].

Vitamin B_2_ content in human milk is affected by the maternal intake of vitamin B_2_ [2,26]. Therefore, the intake of vitamin B_2_ during lactation period should be increased up to 1.7 mg per day [19]. The median vitamin B_2_ intake among the study participants from the first study group was within recommendations, but median intake from the second study period was lower than recommended 1.7 mg per day. Low vitamin B_2_ intake could be related to the fact, that milk and dairy products (a source of vitamin B_2_) are not consumed on daily basis during the lactation period among women in Latvia [15]. Insufficient vitamin B_2_ intake among lactating women has been also reported in France [24].

Fish, meat and meat products are the main sources of vitamins B_3_ and B_6_ [20]. It also seems that both vitamin B_3_ and B_6_ content in human milk is directly affected by the maternal intake of these micronutrients [2,26]. Therefore, women during lactation period should increase their intake of vitamin B_3_ up to 20 mg per day and vitamin B_6_ intake up to 1.5 mg per day [19,20]. Intake of vitamins B_3_ and B_6_ was sufficient among the study participants. Meat and meat product intake probably contributes to the adequate intake of vitamin B_3_ and B_6_, since fish are rarely consumed among lactating women in Latvia [15]. Researchers from Poland [23] also report that intakes of these vitamins among lactating women were sufficient. On the contrary, researchers from Spain, France, and Croatia report insufficient intakes of vitamin B_3_ and B_6_ among post-partum women [11,12,24]. However, none of them provides a further explanation as to why lower vitamin B_3_ and B_6_ intakes are reported.

According to dietary guidelines, women during the gestation period are advised, in addition to food, to consume 400 μg of vitamin B_9_ via dietary supplements to reduce the risk of the foetus developing neural tube defects [29]. No dietary guidelines regarding vitamin B_9_ intake via dietary supplements are currently developed for lactating women in Latvia. Although there are vitamin B_9_ rich food sources such as green vegetables and cereals [20], with food alone it would be challenging to ensure recommended intake of vitamin B_9_ during the lactation period (500 μg per day) [19]. Taking into account that vitamin B_9_ requirements during the lactation period (500 μg per day) are even higher than during the gestation period (400 μg per day) [19], additional vitamin B_9_ intake via dietary supplements should be also recommended for lactating women in Latvia.

According to the data from 72-h food diaries, only 14 participants from the first study period and 17 participants from the second study period were taking dietary supplements containing vitamin B_9_, and the median intake of vitamin B_9_ among the study participants was only about 50% of the recommended intake. Furthermore, researchers from Spain, Poland, France, Croatia and the United States raise a concern that vitamin B_9_ intake among lactating women is insufficient [8,11,12,23,24]. It can lead to an insuffient amount of vitamin B_9_ in human milk and, therefore, insufficient intake of vitamin B_9_ for the breastfed infant [31].

Vitamin B_12_ content in human milk depends on maternal status and intake of this vitamin [32]. The intake of vitamin B_12_ during the lactation period should be 2 μg per day [19]. Vitamin B_12_ is mainly found in animal-based food products [20]. Therefore, women who identify as vegetarians or vegans are advised to additionally consume vitamin B_12_ via dietary supplements [20]. A few participants of this study in the 72-h food diary noted that they are vegetarians, and if vitamin B_12_ dietary supplement was not consumed by them, the median vitamin B_12_ intake did not reach the recommended intake of 2 μg. Apart from vegetarian and vegan participants, the median vitamin B_12_ intake among participating women was sufficient (around 4 μg per day). Similar values regarding vitamin B_12_ intake during the lactation period have been reported by researchers from Poland, France, and the United States [12,23,24].

Overall, this study indicates that the intakes of many essential micronutrients among lactating women were not within the recommendations. A healthy diverse diet can help to ensure the increased requirements during the lactation period for almost all nutrients, but the use of dietary supplements can help when food alone cannot ensure a sufficient amount of micronutrients such as iodine and vitamins B_9_, B_12_, and D [20,29].

However, it should be noted that for some study participants, the total median intake of some micronutrients (calcium, phosphorus, iron, zinc, selenium, vitamin D) was too high and exceeded the tolerable upper intake level. Therefore, dietary supplements should be used with caution [20,29].

Dietary guidelines developed at the national level could potentially ensure that balanced micronutrient intakes are achieved for lactating women in Latvia. The Ministry of Health of the Republic of Latvia are planning to develop dietary guidelines for lactating women in 2022 [33].

## 5. Conclusions

Comparing both study periods, mineral and vitamin intakes among lactating women in Latvia have not changed in recent years (except for increased vitamin D intake in the second study period). Overall, dietary intake of micronutrients such as phosphorus, magnesium, vitamins K, B_3_, B_6_, and B_12_ was mostly sufficient, but intake of calcium, iron, iodine, and vitamins A, D, B_1_, and B_9_ among participants of both study periods did not meet dietary recommendations. Insufficient mineral and vitamin intakes for lactating women could potentially affect the composition of human milk and therefore micronutrient intakes for the breastfed infant. This indicates a need to develop national dietary guidelines in order to improve diets among lactating women in Latvia.

## Figures and Tables

**Figure 1 foods-11-00259-f001:**
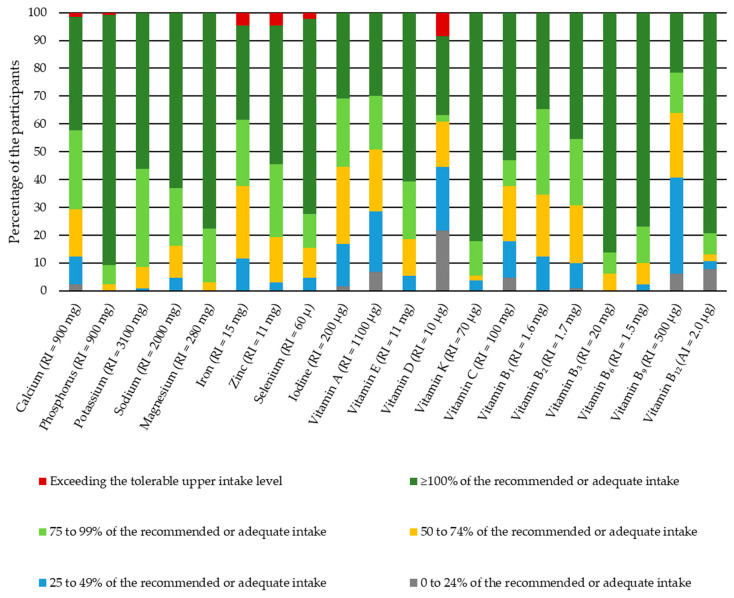
Median micronutrient intakes among the participants, percentage of the recommended intake or adequate intake *n* = 130.

**Table 1 foods-11-00259-t001:** Characteristics of the participants.

Characteristics	First Study Period— November 2016 to December 2017 (*n* = 62)	Second Study Period— January 2020 to December 2020 (*n* = 68)
Maternal characteristics		
Age (years)	31 ± 4 (23–39)	31 ± 5 (23–45)
Maternal body mass index ^1^	22.32 ± 3.19 (17.85–32.18)	23.10 ± 3.58 (18.51–36.57)
Parity	2 ± 1 (1–4)	2 ± 1 (1–5)
Breastfeeding pattern	38—exclusive breastfeeding, 23—partial breastfeeding	48—exclusive breastfeeding, 20—partial breastfeeding
Infant’s characteristics		
Age (months)	5 ± 3 (2–11)	4 ± 3 (1–11)
Sex	31—female, 31—male	34—females, 34—males
Birth weight (kg)	3.51 ± 0.49 (2.39–4.77)	3.62 ± 0.55 (1.63–5.50)
Birth length (cm)	53 ± 3 (47–60)	54 ± 3 (42–61)

^1^ Limitation of the study—maternal body mass index was calculated based on weight and length values given by the participants. No anthropometrical measures were performed during this study.

**Table 2 foods-11-00259-t002:** Foods groups excluded from the diet among the study participants.

Food Group	The Number of Participants from the First Study Period	The Number of Participants from the Second Study Period
Milk and dairy products	*n* = 14	*n* = 10
Cereals containing gluten	*n* = 3	*n* = 1
Eggs	*n* = 2	*n* = 1
Sweets & bakery goods	*n* = 3	*n* = 5
Fish & seafood	*n* = 1	*n* = 1
Soya	*n* = 1	*n* = 0
Nuts	*n* = 1	*n* = 0

**Table 3 foods-11-00259-t003:** Mineral intakes among the participants (*n* = 130).

Nutrient (Unit)	First Study Period (*n* = 62) Median [IQR] ^1^ (Min–Max Values)	Second Study Period (*n* = 68) Median [IQR] ^1^ (Min–Max Values)	Recommended Intake ^2^ for Lactating Women
Calcium (mg)	849.12 [480.09] (165.64–2765.26)	805.03 [476.22] (205.25–1713.05)	900 mg [19,20]
Phosphorus (mg)	1434.99 [632.15] (659.03–3051.46)	1373.11 [420.74] (573.86–2261.63)	900 mg [19,20]
Potassium (mg)	3313.84 [1614.65] (1670.13–12,442.25)	3250.42 [1250.11] (1330.15–5348.34)	3100 mg [19,20]
Sodium (mg)	2297.45 [1192.07] (671.37–4692.19)	2510.82 [1499.50] (650.33–5494.30)	2000 mg [19], 2400 mg [20]
Magnesium (mg)	351.06 [184.93] (210.25–1299.83)	369.91 [163.55] (156.22–891.29)	280 mg [19,20]
Iron (mg)	12.53 [10.50] (4.34–107.13)	12.98 [7.52] (4.41–113.35)	15 mg [19,20]
Zinc (mg)	10.45 [5.01] (4.92–27.99)	11.88 [5.17] (4.12–41.45)	11 mg [19,20]
Selenium (µg)	77.72 [43.86] (16.03–225.57)	88.34 [57.21] (26.91–1493.43)	60 µg [19,20]
Iodine (µg)	163.85 [104.67] (46.47–381.94)	163.92 [100.12] (15.38–356.89)	150–250 µg [19], 200 µg [20]

^1^ Interquartile range. ^2^ The term “recommended intake” (RI) refers to the amount of a nutrient that meets the known requirement and maintains good nutritional status among practically all healthy individuals in a particular life stage [19,20].

**Table 4 foods-11-00259-t004:** Vitamin intakes among the participants (*n* = 130).

Nutrient (Unit)	First Study Period (*n* = 62) Median [IQR] ^1^ (Min–Max Values)	Second Study Period (*n* = 68) Median [IQR] ^1^ (Min–Max Values)	Recommended Intake ^2^ for Lactating Women
Vitamin A, retinol activity equivalents (µg)	868.97 [743.41] (82.35–12,828.07)	759.92 [603.99] (35.63–2540.96)	1100 µg [19,20]
Vitamin E (mg)	12.99 [10.43] (3.90–160.76)	13.12 [8.43] (2.71–49.37)	11 mg [19,20]
Vitamin D (µg)	4.46 [9.46] (0.16–155.53)	6.40 [25.81] (0.00–256.67)	10 mg [19,20]
Vitamin K (µg)	119.19 [84.89] (27.97–1060.79)	116.62 [82.46] (24.33–445.00)	70 mg [21] ^3^
Vitamin C (mg)	114.94 [137.24] (18.37–1015.15)	95.26 [86.88] (0.58–1046.16)	100 mg [19,20]
Vitamin B_1_ (mg)	1.38 [1.11] (0.51–26.43)	1.37 [0.70] (0.43–10.43)	1.6 mg [19,20]
Vitamin B_2_ (mg)	1.70 [1.12] (0.41–26.66)	1.49 [0.93] (0.53–5.88)	1.7 mg [19,20]
Vitamin B_3_ (mg)	29.17 [15.33] (10.18–91.40)	33.45 [14.38] (12.31–69.20)	20 mg [20]
Vitamin B_6_ (mg)	1.91 [1.45] (0.68–21.63)	2.10 [1.38] (0.67–22.71)	1.5 mg [19,20]
Vitamin B_9,_ dietary folate equivalents ^4^ (µg)	269.86 [239.32] (100.72–1822.45)	323.87 [188.43] (95.13–1966.42)	500 µg [19,20]
Vitamin B_12_ (µg)	4.37 [4.78] (0.00–50.59)	4.38 [3.93] (0.00–23.81)	2.0 µg [19], 2.6 µg [20]

^1^ Interquartile range. ^2^ The term “recommended intake” (RI) refers to the amount of a nutrient that meets the known requirement and maintains good nutritional status among practically all healthy individuals in a particular life stage [19,20]. ^3^ “Adequate intake” (AI)—the average nutrient level, based on observations or experiments, that is assumed to be adequate for the population’s needs [21] applies for vitamin K. ^4^ Dietary folate equivalents = food folates (μg) + (1.7 × μg of folic acid from dietary supplements) [21].

**Table 5 foods-11-00259-t005:** Dietary supplement intakes among the study participants.

Nutrient Consumed via Dietary Supplement (Unit)	First Study Period Median (Number of Participants) (Min–Max Values)	Second Study Period Median (Number of Participants) (Min–Max Values)
Calcium (mg)	500.00 (*n* = 11) (120.00–1900.00)	267.00 (*n* = 13) (100.00–600.00)
Phosphorus (mg)	115.00 (*n* = 3) (2.00–1050.00)	not calculated (*n* = 0) (not calculated)
Potassium (mg)	7.00 (*n* = 1) (not calculated)	224.00 (*n* = 2) (50.00–398.00)
Magnesium (mg)	150.00 (*n* = 7) (57.00–155.00)	87.50 (*n* = 16) (28.20–240.00)
Iron (mg)	16.00 (*n* = 15) (8.00–100.00)	16.60 (*n* = 16) (5.00–100.00)
Zinc (mg)	12.50 (*n* = 10) (1.45–18.00)	8.00 (*n* = 20) (1.00–30.00)
Selenium (μg)	30.00 (*n* = 7) (0.10–55.00)	35.00 (*n* = 9) (8.25–200.00)
Iodine (μg)	140.00 (*n* = 7) (140.00–150.00)	150.00 (*n* = 8) (50.00–200.00)
Vitamin A, retinol activity equivalents (µg)	400.00 (*n* = 4) (100.00–1000.00)	721.00 (*n* = 7) (180.00–1600.00)
Vitamin E (mg)	16.00 (*n* = 10) (5.00–150.00)	10.00 (*n* = 17) (3.00–40.00)
Vitamin D (µg)	20.00 (*n* = 21) (3.00–150.00)	50.00 (*n* = 35) (3.00–625.00)
Vitamin K (µg)	70.00 (*n* = 6) (60.00–115.00)	35.00 (*n* = 3) (18.75–37.50)
Vitamin C (mg)	70.00 (*n* = 13) (12.00–950.00)	60.00 (*n* = 19) (8.00–1000.00)
Vitamin B_1_ (mg)	5.00 (*n* = 8) (1.20–25.00)	1.40 (*n* = 11) (0.37–9.00)
Vitamin B_2_ (mg)	2.00 (*n* = 10) (1.12–25.00)	1.50 (*n* = 10) (0.47–5.00)
Vitamin B_3_ (mg)	20.00 (*n* = 8) (3.40–60.00)	17.00 (*n* = 9) (5.33–19.00)
Vitamin B_6_ (mg)	2.60 (*n* = 12) (0.70–20.00)	2.60 (*n* = 15) (1.00–20.00)
Vitamin B_9_, folic acid (µg)	400.00 (*n* = 14) (100.00–1000.00)	400.00 (*n* = 17) (100.00–1000.00)
Vitamin B_12_ (µg)	6.00 (*n* = 12) (2.00–1000.00)	2.50 (*n* = 15) (0.75–500.00)

## Data Availability

The data that support the findings of this study are available on request from the corresponding author [LA]. The data are not publicly available due to containing information that could compromise the privacy of research participants.

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
