# Peer review of "Mineral and Vitamin Intakes of Latvian Women during Lactation Period"

_foods, 2022, doi:10.3390/foods11030259_

Round 1

Reviewer 1 Report

More background information are needed to justify the study.

Please relook into the sentence structure and the paragraph arrangement. Some paragraphs are consisting of 1 sentence only. 

Are there any same/repeat participants in both the study periods?

Are there any significant changes on the dietary pattern of the participants in both the study periods?

There is a high dropped out rate in this study. Appreciate if authors could further justify on this issue.

What is the percentage of contribution from the dietary supplementation towards the overall vitamin and mineral intake among the lactation women in this study? 

What is the reason that contributing to the high supplementation intake among the participants from the second study period?

Discussion could be more structure and organise. More supporting evidences needed.

Table 2 and 3: suggest to include the percentage of achieving the recommendations.

L15: which nutritional guidelines were used?

L22: participants of which study period that did not meet the dietary recommendations?

L104-120: could be presented in table format to increase the clarity

Author Response

We would like to thank the Reviewer for the detailed comments and suggestions provided for the improvement of the manuscript (foods-1520524). We believe that the comments have identified important areas which required improvement. After completion of the suggested edits, the revised manuscript has benefited from an improvement in the overall presentation and clarity. Below, you will find a point-by-point description of how each comment was addressed in the manuscript.  Original comments in boldface, responses in regular typeface.

More background information are needed to justify the study.

We added more background information in the Introduction part.

Please relook into the sentence structure and the paragraph arrangement. Some paragraphs are consisting of 1 sentence only. 

We re-organized the sentence structure and the paragraph arrangement.

Are there any same/repeat participants in both the study periods?

No, there are no repeat participants. We added clarification of it to the text (Line 74).

Are there any significant changes on the dietary pattern of the participants in both the study periods?

No, participants were able to consumed ad libitum diet, no dietary restrictions were applied in both study periods (info provided in Lines 88 to 89).

There is a high dropped out rate in this study. Appreciate if authors could further justify on this issue.

We added explanation for this (Lines 68 to 71).

What is the percentage of contribution from the dietary supplementation towards the overall vitamin and mineral intake among the lactation women in this study? 

We did not calculate the percentage of contribution from the dietary supplements towards the overall vitamin and mineral intake because not all participants were consuming dietary supplements. But in the Table 5 we added information about median and minimal, maximal intakes of nutrients from the dietary supplements among the participants who were consuming dietary supplements during the study period.

What is the reason that contributing to the high supplementation intake among the participants from the second study period?

We added possible explanation for it in the Discussion section (Lines 278 to 287).

Discussion could be more structure and organise. More supporting evidences needed.

We made changes to the Discussion section, by adding more supporting evidence, re-organizing.

Table 2 and 3: suggest to include the percentage of achieving the recommendations.

We decided not to add the percentage of achieving the recommendations in Table 2 and Table 3 because this information is already represented in the Figure 1.

L15: which nutritional guidelines were used?

We added clarification to the Line 15“...to nutritional guidelines at national and European level”.

L22: participants of which study period that did not meet the dietary recommendations?

We added clarification that it applies to both study periods (Line 22).

L104-120: could be presented in table format to increase the clarity

We moved information from the Lines 104 to 120 to the Table 2, to increase the clarity.

Reviewer 2 Report

The authors have examined the Mineral and Vitamin Intakes of Latvian Women During Lactation Period, a topic of timely importance.

 In general, the study confirms several observations. Below I have made some specific points to address for the improvement of the manuscript.

The author should explain;

  • From where these participants were selected (from a clinic, hospital or community etc.)?
  • As this is a cross sectional study, what does the author meant by drop-outs?
  • How the sample size was calculated?
  • Calculation of BMI using the values provided by the women should is a limitation of the study.
  • What does it mean by the numbers given within square brackets in the figure labels?
  • At least simple statistical tests can be used to compare the two groups.
  • Better to present data with percentages, rather showing just only the numbers.
  • Is there any national recommendation for taking supplements during postpartum period? Some of the values were affected by the nutrient values of supplements. As the supplements are taken only by some women these values cannot consider as the overall values for even this studied groups.

Recommend for major revisions to the methodology, results and discussion sections.

Author Response

We would like to thank the Reviewer for the detailed comments and suggestions provided for the improvement of the manuscript (foods-1520524). We believe that the comments have identified important areas which required improvement. After completion of the suggested edits, the revised manuscript has benefited from an improvement in the overall presentation and clarity. Below, you will find a point-by-point description of how each comment was addressed in the manuscript.  Original comments in boldface, responses in regular typeface.

The authors have examined the Mineral and Vitamin Intakes of Latvian Women During Lactation Period, a topic of timely importance.

 In general, the study confirms several observations. Below I have made some specific points to address for the improvement of the manuscript.

The author should explain;

From where these participants were selected (from a clinic, hospital or community etc.)?

We added clarification that “Women were invited to participate in the study via a poster published on social media.” (Lines 63 to 64).

As this is a cross sectional study, what does the author meant by drop-outs?

We added clarification that “...82 women decided not to participate after getting acquainted with the instructions of the study”. (Lines 69 to 70).

How the sample size was calculated?

We added clarification to the text --> “To calculate the minimum sample size for the study period, an online calculator was used. With a probability of 0.05 and a confidence level of 95%, it was calculated that at least 59 participants were needed in each study period” (Lines 64 to 67).

Calculation of BMI using the values provided by the women should is a limitation of the study.

We agree, in the remark of the Table 1 we added note that this is a limitation of the study.

What does it mean by the numbers given within square brackets in the figure labels?

Initially it was references of the information sources, but as suggested from another Reviewer, we switch references in the square brackets to actual recommended intake values shown in the brackets. We hope that it has clarified the information shown in Figure 1.

At least simple statistical tests can be used to compare the two groups.

Yes, we used Kruskal Wallis test to compare both study groups, and found that “No statistically significant differences were found comparing both study groups (p > 0.05).” (Lines 107 to 108) and “no significant differences were found regarding mineral and vitamin intakes comparing both study periods (p > 0.05), except for vitamin D intake, which was higher among the participants from the second study period (p = 0.02).” (Lines 142 to 144).

Better to present data with percentages, rather showing just only the numbers.

Percentages of nutrient intake values are shown in Figure 1.

Is there any national recommendation for taking supplements during postpartum period? Some of the values were affected by the nutrient values of supplements. As the supplements are taken only by some women these values cannot consider as the overall values for even this studied groups.

There are only national recommendations regarding D vitamin intake during postpartum period. These recommendations are included in the national nutritional guidelines for women during gestation period. Separately currently there are no nutrition guidelines for women during lactation period in Latvia.

Recommendations regarding vitamin D intake via dietary supplements during lactation period was added to the text (Lines 270 to 272).

Recommend for major revisions to the methodology, results and discussion sections.

After completion of the suggested edits from all three Reviewers, we believe that the revised manuscript has benefited from an improvement in the overall presentation and clarity.

Reviewer 3 Report

The study addresses the important issue of the quality of diet of breastfeeding women. It is worth noting that in the literature there is much less research on the diet of lactating women than of pregnant women.

Minor remarks.

In Materials and Methods, the sentence "Exclusion criteria for the study were noncompliance with the inclusion requirements  and/or unsigned consent blank" (lines 72-73) is not needed, because this information is obvious in the light of the inclusion criteria (lines 64-71)

In the Results in Table 2 with regard to calcium, it is not known what the numbers 2 and 3 mean in superscript in the fourth column.

The method of specifying median ± interquartile ranges instead of median with interquartile ranges (p25-p75) is not clear.

What exactly does the Recommended daily intake in Tables 2 and 3 mean. Are these Recommended Dietary Allowances (RDA) or Adequate Intake (AI) values? This is relevant as the authors write “Overall, median phosphorus, magnesium, vitamins K, B3, B6, B12 intake among the participants was adequate (the recommended daily intake was reached for at least 75% of the participants)”

This table should also be technically corrected, because the given values, for example, for iron, zinc in the second and third columns do not coincide with the first column

In tables 3, I suggest using the name niacin instead of vitamin B3 and folic acid instead of vitamin B9

In figure 1, on the horizontal axis, instead of the reference number, I propose to put the exact values ​​of the recommended intake with the diet with which the intake of these components by mothers was compared. Iodine is a good example to justify this approach, because the article shows that once the authors used the value of 200 μg (line 144), and once the value of 150 μg (line 221)

In the sentence “The intake of many vitamins (A, D, B1, B2, B9) among the participants did not reach the recommended daily intake for lactating women (Table 3)”, the word “most” should be added before participants.

In the sentence “For more than half of the participants…” (line 154) it would be better to write “Over 60% of participants …” because that would be according to Figure 1

In the sentence “D vitamin was the most often consumed dietary supplement (n = 57), followed by vitamin C (n = 33) and iron (n = 31), vitamin B9 (n = 31) (Table 4)” it would be worth adding that it is about both study periods.

The sentence  “Information regarding nutrient intakes via dietary supplements are compiled in Table 4.” is misleading, as this table does not give the amount of nutrient intake from supplements, but only the number of women using dietary supplements.

It is worth noting that the information on what amount of nutrients was provided by the traditional diet and what amount by supplements would be very interesting.

Discussion

How the authors would address the problem of dietary iron and low meat consumption (lines 198-202). As is known in recent years, in Western countries it is suggested to limit red meat (to 500 g per week) and eat such meat no more than 2-3 times a week. With such recommendations, it is obvious that the consumption of well-absorbed iron will not meet the needs. How to combine these two issues?

Current RDA values for iron for breastfeeding women in Poland should be provided (line 207) (instead of 7 mg, now is 10 mg)  https://ncez.pzh.gov.pl/abc-zywienia/normy-zywienia-2020/

In the sentence "However, it should be noted that calculated values regarding selenium intake in this study were not only from food but also from dietary supplements" (lines 218-220), instead of "this study", I propose to write "our study" because it is not known exactly what research this information concerns.

How the authors would explain the low intake of vitamin D from supplements that were used by 57 women (“Although almost 50% of the study participants noted that they use dietary supplements containing D vitamin, the median vitamin D intake among participating women was only around 5 to 6 μg per day, which is half of the recommended daily intake (10 μg per day)”

I wonder if in the sentence “However, previous data regarding habitual food intake among lactating women in Latvia report seldom consumption of cereals & cereal products but milk & dairy products consumption - twice a week”  instead of “but” should be “and”?

Regarding the information “However, it should be noted that for some study participants, the total median intake of some nutrients (calcium, phosphorus, iron, zinc, selenium, vitamin D) was too high and exceeded the tolerable upper intake level. Therefore dietary supplements should be used with caution. Daily nutrient intakes above tolerable upper level pose a risk to both the mother's and the breastfed infant's health "  it should be taken into account that the period of breastfeeding is relatively short, and UL has a large safety margin.

Author Response

We would like to thank the reviewer for the detailed comments and suggestions provided for the improvement of the manuscript (foods-1520524). We believe that the comments have identified important areas which required improvement. After completion of the suggested edits, the revised manuscript has benefited from an improvement in the overall presentation and clarity. Below, you will find a point-by-point description of how each comment was addressed in the manuscript.  Original comments in boldface, responses in regular typeface.

The study addresses the important issue of the quality of diet of breastfeeding women. It is worth noting that in the literature there is much less research on the diet of lactating women than of pregnant women.

Minor remarks.

In Materials and Methods, the sentence "Exclusion criteria for the study were noncompliance with the inclusion requirements  and/or unsigned consent blank" (lines 72-73) is not needed, because this information is obvious in the light of the inclusion criteria (lines 64-71)

We agree with the Reviewer’s comment, and we have removed this sentence.

In the Results in Table 2 with regard to calcium, it is not known what the numbers 2 and 3 mean in superscript in the fourth column.

Numbers 2 and 3 were corrected to appropriate references – [17,18].

The method of specifying median ± interquartile ranges instead of median with interquartile ranges (p25-p75) is not clear.

We corrected “median ± IQR” to “Median [IQR]” in Tables 2 and 3.

What exactly does the Recommended daily intake in Tables 2 and 3 mean. Are these Recommended Dietary Allowances (RDA) or Adequate Intake (AI) values? This is relevant as the authors write “Overall, median phosphorus, magnesium, vitamins K, B3, B6, B12 intake among the participants was adequate (the recommended daily intake was reached for at least 75% of the participants)”

Latvian nutritional guidelines (Ieteicamās enerģijas un uzturvielu devas Latvijas iedzīvotājiem. Veselības ministrija, 2017. gads) are based on Nordic Nutrition Recommendations, where the term “recommended intake” is used.  The term refers to the amount of a nutrient that meets the known requirement and maintains good nutritional status among practically all healthy individuals in a particular life stage.

We added the explanation of the term “recommended intake” as remark under the Tables 2 and 3.

Except for vitamin K, where only “adequate intake” value is reported (added this clarification and explanation of the term as the remark under Table 3).

And in the text word “adequate” was replaced with “recommended intake” where necessary.

This table should also be technically corrected, because the given values, for example, for iron, zinc in the second and third columns do not coincide with the first column

We technically corrected the table 3 so the given values coincide the first column.

In tables 3, I suggest using the name niacin instead of vitamin B3 and folic acid instead of vitamin B9

We decided to leave the names vitamins B3 and B9, but in the table 4 we added clarification that regarding vitamin B9 values, we have calculated dietary folate equivalents to include both intake of folates via foods and folic acid via dietary supplements.

In figure 1, on the horizontal axis, instead of the reference number, I propose to put the exact values ​​of the recommended intake with the diet with which the intake of these components by mothers was compared. Iodine is a good example to justify this approach, because the article shows that once the authors used the value of 200 μg (line 144), and once the value of 150 μg (line 221)

We made above-mentioned suggestions in Figure 1 (exact value of recommended intake or adequate intake put on the horizontal axis for each nutrient).

In the sentence “The intake of many vitamins (A, D, B1, B2, B9) among the participants did not reach the recommended daily intake for lactating women (Table 3)”, the word “most” should be added before participants.

In above-mentioned sentence the word “most” was added before the word “participants”.

In the sentence “For more than half of the participants…” (line 154) it would be better to write “Over 60% of participants …” because that would be according to Figure 1

We checked again the Figure 1. It would be more correct to say 40% of the participants, not 60%. To clarify, we have rewritten this sentence --> “For ~40% of the participants, the median vitamin B9 intake did not even reach half of the recommended intake (Figure 1)”.

In the sentence “D vitamin was the most often consumed dietary supplement (n = 57), followed by vitamin C (n = 33) and iron (n = 31), vitamin B9 (n = 31) (Table 4)” it would be worth adding that it is about both study periods.

We added text that it applies to both study periods.

The sentence  “Information regarding nutrient intakes via dietary supplements are compiled in Table 4.” is misleading, as this table does not give the amount of nutrient intake from supplements, but only the number of women using dietary supplements.

It is worth noting that the information on what amount of nutrients was provided by the traditional diet and what amount by supplements would be very interesting.

We added information regarding nutrient intakes provided via supplements in the Table 5. Therefore, we believe that the name of the Table can stay as it was previous “Information regarding nutrient intakes via dietary supplements”.

Discussion

How the authors would address the problem of dietary iron and low meat consumption (lines 198-202). As is known in recent years, in Western countries it is suggested to limit red meat (to 500 g per week) and eat such meat no more than 2-3 times a week. With such recommendations, it is obvious that the consumption of well-absorbed iron will not meet the needs. How to combine these two issues?

We agree that to limit red meat intake and to ensure sufficient iron intake is challenging, but we tried to combine these two issues in the text (Lines 211 to 221).

Current RDA values for iron for breastfeeding women in Poland should be provided (line 207) (instead of 7 mg, now is 10 mg)  https://ncez.pzh.gov.pl/abc-zywienia/normy-zywienia-2020/

We made necessary changes in the text (Line 228) and added this information source also to the references.

In the sentence "However, it should be noted that calculated values regarding selenium intake in this study were not only from food but also from dietary supplements" (lines 218-220), instead of "this study", I propose to write "our study" because it is not known exactly what research this information concerns.

We changed “this study” to “our study”.

How the authors would explain the low intake of vitamin D from supplements that were used by 57 women (“Although almost 50% of the study participants noted that they use dietary supplements containing D vitamin, the median vitamin D intake among participating women was only around 5 to 6 μg per day, which is half of the recommended daily intake (10 μg per day)”

We added reference to the national nutritional guidelines for women during gestation period, where it is stated that women during gestation and lactation period should additional consume 10 μg of vitamin per day (20 μg during winter period). And that obtained data of our study indicate that it should be more emphasized around women in Latvia that additional vitamin D via dietary supplements during lactation period is recommended (Lines 270 to 287).

I wonder if in the sentence “However, previous data regarding habitual food intake among lactating women in Latvia report seldom consumption of cereals & cereal products but milk & dairy products consumption - twice a week”  instead of “but” should be “and”?

In this sentence, we changed “but” to “and”.

Regarding the information “However, it should be noted that for some study participants, the total median intake of some nutrients (calcium, phosphorus, iron, zinc, selenium, vitamin D) was too high and exceeded the tolerable upper intake level. Therefore dietary supplements should be used with caution. Daily nutrient intakes above tolerable upper level pose a risk to both the mother's and the breastfed infant's health "  it should be taken into account that the period of breastfeeding is relatively short, and UL has a large safety margin.

We agree, that breastfeeding period is relatively short period of time. Therefore, we removed the sentence “Daily nutrient intakes above tolerable upper level pose a risk to both the mother's and the breastfed infant's health” from the text.

Round 2

Reviewer 1 Report

Background needs to be improved further to justify the importance of this study.

The authors did not justify on the high dropped out rate from this study. 

Discussion could be more structure and organise by providing more supporting evidences.

Author Response

We would like to thank the reviewer for second suggestions provided for the improvement of the manuscript (foods-1520524). Below, you will find a point-by-point description of how each comment was addressed in the manuscript.  Original comments in boldface, responses in regular typeface.

Background needs to be improved further to justify the importance of this study.

We added additional information in the Introduction part to justify the importance of the study (Lines 53 to 65).

The authors did not justify on the high dropped out rate from this study. 

We added further explanation - "82 women decided not to participate after getting acquainted with the instructions of the study or were unable to collect sufficient amount of human milk for the study (data regarding human milk composition among lactating women in Latvia reported in articles Aumeistere et al., 2019 [15] and Aumeistere et al., 2021 [17]). (Lines 82 to 86).

Discussion could be more structure and organise by providing more supporting evidences.

We added additional information in the Discussion part to provide more supporting evidence. Structure and organisation of the Discussion section was improved (Lines 196 to 199, 204 to 208, 213 to 218, 224 to 225, 229 to 231, 235 to 253, 275 to 277, 284 to 287, 290 to 292, 294 to 296, 309 to 310, 314 to 316, 321 to 325, 333 to 335, 356 to 360, 365 to 367, 374 to 376, 382 to 389, 395 to 396, 412 to 413, 433 to 435).

Reviewer 2 Report

The authors have substantially improved the manuscript after accommodating the reviewers' comments. The authors satisfactorily addressed the comments given by me.

Thank you

Author Response

As the Reviewer 2 have not provided further comments about the manuscripts, no additional changes have been made, except for the improvements in the Research Design. Additional improvements were suggested by the Reviewer 1, so they are included in the revised manuscript which can be evaluated by the Reviewer 2. Thank you.